# A community developed conceptual model for reducing long-term health problems in children with intellectual disability in India

**Manisha Nair** [1]*, **Mythili Hazarika**[2◉], **M Thomas Kishore**[3◉], **Nabarun Sengupta**[4], **Ganesh Sheregar**[5], **Hayley MacGregor**[6], **Mary Wickenden**[6], **Neel Harit Kaushik**[7], **Prarthana Saikia**[8], **Maureen Kelley**[9], **Sasha Shepperd**[1]

1 Nuffield Department of Population Health, University of Oxford, Oxford, United Kingdom, 2 Gauhati Medical College and Hospital, Guwahati, Assam, India, 3 National Institute of Mental Health and Neurosciences, Bengaluru, India, 4 Meghalaya Parents Association for Disabled, Shillong, India, 5 National Institute for the Empowerment of Persons with Intellectual Disabilities, Secunderabad, India, 6 Institute of Development Studies, University of Sussex, Brighton, United Kingdom, 7 Assam Don Bosco University, Sonapur, Assam, India, 8 Indian Institute of Technology, Guwahati, Assam, India, 9 Center for Ethics in Health Care and Department of Internal Medicine, Oregon Health & Science University; Portland, Oregon, United States of America

◉ These authors contributed equally to this work.
* manisha.nair@npeu.ox.ac.uk

**Data Availability Statement:** Our qualitative data includes transcripts from a public engagement and involvement work. Even if we remove the names and other personal details, the data would be still

## Abstract

Children with intellectual disability (ID) have a higher risk of long-term health problems in adulthood. India has the highest prevalence of ID of any country with 1.6 million under-five children living with the condition. Despite this, compared with other children, this neglected population is excluded from mainstream disease prevention and health promotion programmes. Our objective was to develop an evidence-based conceptual framework for a needs-based inclusive intervention to reduce the risk of communicable and non-communicable diseases among children with ID in India. From April through to July 2020 we undertook community engagement and involvement activities in ten States in India using a community-based participatory approach, guided by the bio-psycho-social model. We adapted the five steps recommended for the design and evaluation of a public participation process for the health sector. Seventy stakeholders from ten States contributed to the project: 44 parents and 26 professionals who work with people with ID. We mapped the outputs from two rounds of stakeholder consultations with evidence from systematic reviews to develop a conceptual framework that underpins an approach to develop a cross-sectoral family-centred needs-based inclusive intervention to improve health outcomes for children with ID. A working Theory of Change model delineates a pathway that reflected the priorities of the target population. We discussed the models during a third round of consultations to identify limitations, relevance of the concepts, structural and social barriers that could influence acceptability and adherence, success criteria, and integration with existing health system and service delivery. There are currently no health promotion programmes focusing on children with ID in India despite the population being at a higher risk of developing comorbid health problems. Therefore, an urgent next step is to test the conceptual model to determine acceptance and

potentially identifiable of the persons/organisations including government representatives whose views might be considered professionally and politically sensitive. This is particularly true for our work that focuses on intellectual disability and includes quotes related to stigma and social injustice. So, to protect confidentiality we did not seek consent from the participants to share data in an open access repository. Participants consented to including quotes in scientific reports, publications and presentations, but not to have entire transcripts made publicly available or available to a third party. Thus, making entire transcripts publicly available would breach compliance with participant consent. We would be happy to consider requests for data from interested individuals or organisations provided we could work out ways to share non-identifiable information. Information on how to access data is available on Oxford Population Health Department's website (https://www.ndph.ox.ac.uk/data-access) and requests can also be emailed directly to data.access@ndph.ox.ac.uk. Oxford Population Health's Data Repository holds data from multiple studies and enables long-term storage and availability to maximise the value of these data for public health. Requests can be made to access raw data, summary tables and analyses that have not been released in publications or through online repositories.

**Funding:** The CEI work included in this paper was funded by a National Institute for Health and Care Research (NIHR), RIGHT Call 3 Proposal and Partnership Development Award (Ref: NIHR201720 to MN and MH). The work was also supported by a Medical Research Council Transition Support Award (Ref: MR/W029294/1 to MN). The funders had no role in the study design, data collection, analysis, or writing the paper.

**Competing interests:** The authors have declared that no competing interests exist.

effectiveness within the context of socio-economic challenges faced by the children and their families in the country.

## Introduction

People with intellectual disability (ID) are at an increased risk of premature ageing and death [1–3]. The adverse impact of risk factors such as malnutrition, infection, stress and anxiety, can be far reaching, leading to poor health outcomes, a lower quality of life and higher health-care costs for people with this condition [4]. ID is "a group of etiologically diverse conditions originating during the developmental period characterised by significantly below average intellectual functioning and adaptive behaviour" [5]. Of all developmental disabilities, this is the largest contributor to years-lived-with-disability [6]. Children with ID have a 70–98% higher risk of multiple long-term health problems in adulthood [7].

India has the highest rate of ID of any country [6], a range of 1%-6.3% [8–11] is reported, compared with 0.22–1.55% globally [12]. Applying a rate of 1% [8–11] to population figures from Census 2011 [13], the total population living with ID would be at least 12 million including about 1.6 million under-five children. Prevalence is higher in some States such as the Northeast States [10]. As life expectancy of people with ID continues to increase in India [14] and in other low-and-middle income countries (LMICs) [15], preventing and reducing modifiable risk factors for long-term health problems from an early age is important to ensure healthy ageing and to prevent premature death. Evidence suggests that compared with other children, persons with ID are poor [16] and often neglected in mainstream disease prevention and health promotion programmes [2]. They are disproportionately at higher risk of developing multiple health problems due to lack of physical activity, unhealthy diet, social exclusion, poverty and the adverse effects of prolonged pharmacological treatments for co-morbidities such as epilepsy [3, 7, 17–19]. Our objective was to develop an evidence-based conceptual framework for a needs-based inclusive intervention through community consultations to reduce the risk of long-term health problems among children with ID in India.

## Methods

### Ethics statement

The Community engagement and involvement (CEI) activities reported in this paper were mainly consultations with key stakeholders to develop an evidence-based conceptual frame-work designed to improve services. The community consultations were undertaken within the recommended ethical standards, including verbal consent to use quotes from stakeholders without identifying them, and their right to withdraw this approval any time without providing any reason. This type of CEI activity is exempt from research ethics review by the University of Oxford's ethics committees and by the funder, National Institute for Health and Care Research (see INVOLVE statement https://www.invo.org.uk/posttypepublication/public-involvement-in-research-and-research-ethics-committee-review/). Stakeholders contributed to the design of the conceptual framework, and were not involved as research participants. This work was also motivated by an ethical commitment to engaging communities in study design to ensure that the study plan reflects the priorities of the community and is responsive to the specific community context, including contextual understandings of the potential risks and benefits of research.

From April through to July 2020 we undertook community engagement and involvement (CEI) activities using a community-based participatory approach, guided by the bio-psycho-social model. This approach underpins the World Health Organisation's (WHO's) International Classification of Functioning Disability and Health (ICF) [20] to focus on behaviour change factors and access to healthcare services which could influence opportunities, capability and motivation of parents and children to engage with a health related intervention designed to reduce risk factors for poor health outcomes [21,22]. Working with stakeholders we developed a conceptual framework and a working model of a 'Theory of Change' to delineate a pathway that reflected the priorities of the target population [23]. We adapted the five steps recommended for the design and evaluation of a public participation process for the health sector [24].

## 1. Representation of a range of community stakeholders

We consulted with 70 key community stakeholders in ten States in India, eight in the northeast region that have a high prevalence of ID (Assam, Arunachal Pradesh, Manipur, Meghalaya, Nagaland, Sikkim, Mizoram, Tripura) and Delhi and Maharashtra. We aimed to include stakeholders from all socio-economic backgrounds. There were no exclusion criteria as we wanted to include a wide and diverse range of views from people who either had children with intellectual disability or had an experience of working with or supporting children/adults with intellectual disability. Stakeholders were identified through snowballing, starting with a local organisation of parents of children with ID, and initiated by co-investigators from India (psychologist MH; and parents' representative NS). The stakeholders were: i) parents of children with ID; ii) professionals from health, education and social care who were engaged in caring for children and adults with ID; iii) policy makers (including commissioners), lawyers, and members of parents' organisation and other non-governmental organisations (NGOs) working with people with ID.

## 2. A systematic approach to stakeholder interviews and consultation

We conducted three iterative rounds of one-to-one consultations with each of the 70 stakeholders. To overcome the constraints of social restrictions due to the COVID-19 pandemic, we followed guidance on e-consultations [25] to enable us to extend opportunities to as many stakeholders as possible via emails, and through tele and video-conferencing. We organised face-to-face meetings (mainly with parents) after the lockdown measures were eased in June 2020. Two psychologists (MH and NHK), parents' representative (NS) and a PhD student (PS) from India conducted the consultations.

In the first round of consultations, we used five case studies of children with ID to initiate discussions. The case studies were compiled through an in-depth face-to-face discussion with parents prior to the COVID-19 pandemic. They provided stakeholders with a general understanding of the day-to-day living experiences, challenges, health and other concerns, fears and hopes of parents/carers of people with ID. We explored broader and more general perspectives on the:

- Day-to-day problems and needs of children with ID and their families.

- Support systems and structures available to the children and their families.

- Concerns about health and wellbeing of the children (current and future).

In round-2, we used a semi-structured topic guide to explore the stakeholders' perception of risk factors for communicable and non-communicable diseases in children with ID and

how we could start to address them within the wider context of the structural and social barriers. In round-3, we obtained critical feedback on a draft conceptual framework (described below) and a Theory of Change model developed from the findings of round 1 and 2. In particular, the stakeholders were asked to identify limitations, assess the relevance of the concepts, acceptability, success criteria, integration with existing care pathways and health system (private and public), how these relate to existing health policies and who would be best placed to deliver the intervention.

### 3. Information generated and iteratively reviewed

Consultations undertaken in local languages were transcribed and translated to English. Transcripts of each of the 70 consultations were coded, using set and emergent themes with constant comparison [26]. The themes were assessed by study investigators (MN, SS, MH, NS, HM, MW, TKM), and ambiguous and uncertain themes were further explored with stakeholders in subsequent rounds. An iterative team coding approach was used to check interpretations and improve validity [27].

### 4. Developing a conceptual framework and theory of change

We mapped the findings from the consultations with evidence from systematic reviews to develop a conceptual framework to inform the development of a health promotion and disease prevention intervention that could reduce the known risk factors for long-term health problems in children with ID while taking into account the broader socio-economic contexts and barriers. The framework was co-developed by the study team from India and the UK working with experts from the Indian government and non-government institutions including three national bodies:

1. National Institute for the Empowerment of Persons with Intellectual Disabilities (NIEPID), Secunderabad under the Ministry of Social Justice and Empowerment, Government of India;

2. National Institute of Mental Health and Neurosciences (NIMHANS), Bengaluru under the Ministry of Health and Family Welfare, Government of India; and

3. PARIVAAR–National Confederation of Parents' Organisations (NCPO) working with persons with ID.

Using our findings we worked backwards to sketch out the pathways for potential change and its preconditions, incorporating the contextual, structural and social shifts that could facilitate the intended changes within a Theory of Change.

### 5. Triangulation of findings for decision-making

Information from the three rounds of inter-linked and iterative discussions were triangulated to modify and refine the conceptual framework for a cross-sectoral family-centred intervention for children with ID and their parents. This was undertaken by the study team working with the same set of stakeholders who developed the initial conceptual framework (step 4 above).

## Results

Despite the constraints of lockdown and social distancing regulations due to the COVID-19 pandemic in India, we achieved a good representation of parents (n = 44) and other stakeholders working with children and adults with ID (n = 26) from the ten States (Table 1).

**Table 1. Characteristics of stakeholders who participated in the three rounds of consultations.**

| Stakeholders and characteristics | Number |
|---|---|
| **Parents (total)** | **44** |
| Place of residence | |
| Living in rural areas | 20 |
| Living in urban areas | 24 |
| Socio-economic status | |
| Below poverty line* | 15 |
| Above poverty line | 29 |
| Education | |
| Primary and/ or secondary education | 17 |
| Higher education/ professional | 27 |
| **Other stakeholders (total)** | **26** |
| Teachers/ educators | 5 |
| Social workers | 2 |
| Occupational and rehabilitation therapist | 1 |
| Psychologists | 8 |
| Psychiatrists | 8 |
| Policy advocates including members of civil society organisations | 2 |

*The Planning Commission of India defines 'Below Poverty Line' households as households (average five family members) with per capita consumption expenditure of INR 672.8 on a monthly basis in rural areas and INR 859.6 in urban areas at prices prevailing in 2009–10 (Planning Commission; Government of India, 2012)

## Round-1 outputs

Three key themes emerged from the first round of consultation, the most important being the 'future of the children', followed by 'inclusion' and 'support systems'. Almost all parents were concerned about the future care and wellbeing of their children when they would no longer be able to provide care. Marginalisation and exclusion were the factors driving insecurity and exacerbated their worries about the future of their child.

> *"I am a single parent, after my son's birth my husband divorced me and so we are staying with my parents. But I am suffering with insecurity when I think about his future life after my demise or in my old age. Please if it is possible you should include this issue in you[r] programme."*

> *(Mother of a child with ID)*

> *"People should behave as normally as they behave with anyone else. That is the bare minimum they can do. Moreover, I expect society to be more tolerant and accommodating in general."*

> *(Mother of a 43 year old woman with ID)*

The majority of professionals (educators, health and social care workers) and members of parents' organisations felt that parents should be provided with adequate knowledge and information, and be actively involved in educating and caring for the child at home. Active involvement and regular communication with healthcare providers and educators could reduce their anxiety, improve their understanding and acceptance of the child's condition, and help to

improve their mental wellbeing through support. One of the project stakeholders believed that parents are not adequately informed about their children's condition and provided suggestions on how support can be provided at home.

*". . .they put the entire hope and aspiration on the schools and SSA [Sarva Shiksha Abhiyan] functionaries. . . . . . ..Empowering teachers with the right skills is necessary and also empowering parents about their child's condition and how to care at home is equally important."*

*[Special educator, SSA (Inclusive education programme)]*

Improving access to and involvement with healthcare services in both the public and private sectors were identified as important to improve the support system for children with ID and improve physical and mental health of the children and their parents. The overarching concern of parents about securing the future for the child beyond just putting food on the table and day-to-day care was an important observation.

*"The benefit of parental and family involvement in the rehabilitation program has never been understood better than now. Professionals are beginning to realise the benefit of family approaches over merely child-centred ones. This point requires consideration."*

*(Member of a parents' organisation)*

### Round-2 outputs

A few professionals were aware of a higher risk of communicable and non-communicable diseases in people with ID, and suggested that there was a need to improve awareness about the importance of early interventions to reduce future risk of diabetes, cardiovascular diseases, mental health problems, tuberculosis, COVID-19 and other infections. An increased risk of physical health problems was attributed to a range of factors that included malnutrition, infection, poverty, poor access to healthcare services, societal factors such as those related to stigma and low inclusion, behavioural and lifestyle factors, and a lack of awareness and knowledge about health promotion and disease prevention among parents and individuals with ID.

*"Children and persons with ID besides being faced with known health concerns, are more vulnerable and susceptible to other illnesses due to lower immunity levels and also because they are not aware and equipped with required knowledge to take good care of their general health–both mental and physical."*

*(Deputy Director, Civil Society organisation)*

*"Another critical concern is the lack of acknowledgment among parents and other caregivers about the sexual needs, identities and expressions of children and adolescents with ID. Championing for the same could be viewed with a lens of censorship, when societal norms dictate that these are 'private' matters. As a result, these are not addressed timely and adequately, thereby impacting the mental well-being of people with ID and sometimes also resulting in unwanted changes in their behaviours like resorting to violence and so on."*

*(Deputy Director, Civil Society organisation)*

Parents of children with ID mostly reported not being aware of any increased risk of long-term health problems, but mentioned that they would do everything required to prevent it.

The statements quoted above also point to a lack of awareness of inclusive approaches amongst parents, professionals and the wider society. An educator raised concern about the lack of health promotion and disease prevention programmes for children with ID.

> *"Government has taken up education of these children as a priority, whereas health [promotion] still remains to be addressed. Maybe it will be prioritised in the next phase. NGOs are also focusing more on education than health. I would press for a health related programme."*

> *(Educator, inclusive education programme)*

The common recommendations for reducing the increased risk of long-term health problems were to increase physical activity levels in children with ID by integrating this with daily activities and to improve their nutrition. Social workers supporting children with ID suggested that the intervention should be tailored to the needs of children from different socio-economic backgrounds, and take into account the severity of disability and any co-existing conditions. We were advised to keep the intervention simple, with separate components for undernutrition and obesity, to ensure it is sustainable and build on the existing Early Intervention Services for children 0–3 years offered by the Rashtriya Bal Swasthya Karyakram of the Ministry of Health and Family Welfare, Government of India [28]. This will also facilitate an inclusive approach, as the intervention will be embedded in a mainstream programme for all children. Experts from the three national bodies advised us to target the age group of 4–10 years to support continuity of care and integration with the existing policies and programmes.

> *"Early childhood interventions are necessary and should be continued as part of the programme. If a child does not know how to brush, how can you improve dental hygiene?"*

> *(President, National parents' organisation)*

## Conceptual framework that underpins an intervention to reduce risk factors for long-term health problems

A report on the state-level burden of disease in the general population highlights the triple-burden of infectious and lifestyle related diseases, and mental health problems among the adult population in India. These are associated with risk factors such as malnutrition, high blood pressure, high blood sugar, stress and anxiety, which are in turn related to unhealthy diet, tobacco, alcohol and drug use, socio-economic factors, and unsafe drinking water, sanitation, and hygiene [29]. A review of systematic reviews indicated that interventions to increase physical activity had a large beneficial effect on physical and psychosocial health moderated by age and level of disability [30] although this evidence mainly came from high-income countries. Sustained behaviour change was higher when combined with a healthy lifestyle and supportive structural and socio-economic environments [31]. Community-based programmes that provided meaningful parent participation (e.g. through having fun, role models) reduced the burden of caregiving [32]. Integrated mental health counselling services within a health promotion programme for children with ID was found to be effective in reducing stress and anxiety among parents [33]. Evidence also supported collaboration with adults with ID in all aspects of research to improve the implementation and sustainability of programmes [34].

Combining the evidence with the findings from the stakeholder consultations, we developed a conceptual framework that covered an enabling environment, structural and socio-economic determinants, integration with health and social care, intermediate modifiable risk

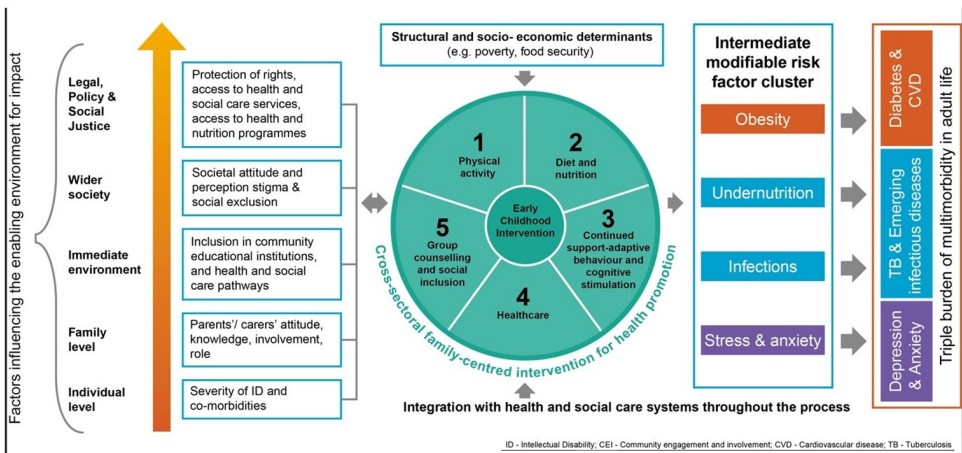

**Fig 1. Conceptual framework for a community-based intervention for preventing and reducing risk of long-term health problems in children with intellectual disability.**

factors and the long-term burden of multiple health problems in later life (Fig 1). This led to the identification of five evidence-based core components to underpin a family-centred, cross-sectoral, community-based intervention for children with ID aged 4 to 10 years:

1. A physical activity programme for the children.

2. A guide to a healthy diet to provide good nutrition to the children using locally available and affordable food.

3. Training parents to continue regular adaptive behavioural and cognitive stimulation, including in everyday activities alongside other children who do not require specialist resources.

4. Education and training to improve parent's knowledge of accessing healthcare services.

5. Group counselling for parents and children to increase their capability to address behavioural issues, individual barriers to exercise and how to follow a healthy diet, improve emotional and mental wellbeing, and to encourage social inclusion.

In consultation with stakeholders we developed a Theory of Change (Fig 2) to identify how the community-based intervention leads to a lower risk for longer-term health problems in our target population of children with intellectual disability, and the necessary preconditions for a beneficial change to occur for these children and their parents. The Theory of Change model therefore includes several stages starting from how the invention was conceptualised during the proposal and partnership development phase to what long-term impact would look like if the intervention was developed and tested effectively leading to the expected outcomes, and the necessary pre-conditions for the success of the intervention were met. The model shows that the cross-sectoral family-centred programme is not a standalone intervention, but will need to be supported by an inclusive enabling environment (Fig 1). Therefore, it will be essential to identify and mitigate social and health system barriers, and actively promote community mobilisation to integrate the intervention within existing health and social care services (Fig 2). While the intervention could achieve the immediate outcomes within a short period (example improving physical activity, diet, and access to healthcare, and reducing stress and anxiety), it will take longer to observe an impact on reducing the long-term health

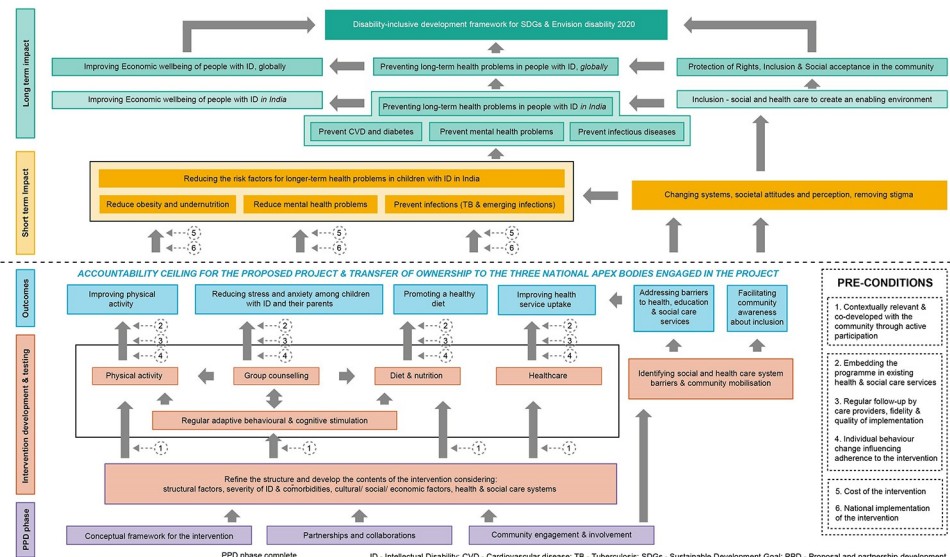

**Fig 2. Theory of change model.**

problems in children with ID. This is clearly acknowledged by the accountability ceiling and the plan to transfer ownership of the intervention to the three national apex bodies engaged in developing the conceptual models. The Theory of Change model will evolve as the intervention is tested and implemented, along with growing partnerships and changes in political will, which will have an effect on the accountability ceiling and transfer of ownership from local to the national bodies. The stakeholders acknowledged that widespread advocacy would be required at each stage of the Theory of Change model to ensure that the intervention tested locally is appropriately adopted at a national level to have the desired impact on policies and practices for a sustainable impact. The team developing the intervention is perfectly placed to undertake such advocacy work as it includes representatives of three national organisations that are involved in developing national policies and programmes for people living with intellectual disability, as well as community stakeholders, people with intellectual disability, and parents of children with intellectual disability.

## Round-3 outputs

Stakeholders voiced strong support for the conceptual framework that inform the intervention to guide the planning and delivery of services for this population, and agreed that *"it will mark a beginning" (Educator, Inclusive education programme)* and promote the *"value of life of the child" (Carer of a child with ID).*

> *"The programme would be a good start of a holistic intervention. Done with perseverance, it can positively impact the child's coping skills, health and well-being and relieve stress in the family as the child begins to benefit from inputs received."*
>
> *(Deputy Director, Civil Society organisation)*

Importantly, it was suggested that we develop a contextually relevant intervention and an enabling environment that will address the challenges and potential barriers to adherence to the intervention components.

While several stakeholders suggested that grass-roots level public sector health and nutritional workers such as Anganwadi workers (village nutrition worker) and community health workers (Accredited Social Health Activist (ASHA)) should be trained to deliver the intervention, others were worried about the negative impact on quality and sustainability if it is delivered by this already over-burdened workforce. It was felt that care providers employed by government and non-government institutions, such as the health system, nutrition and social welfare sectors and schools for children with disabilities would be better placed to deliver the intervention due to their understanding of the children's complex needs. The district disability rehabilitation centres [35] could coordinate outreach, delivery of the intervention and follow-up. The trained workers in these centres are already working towards improving societal awareness and attitudes, community inclusion of people with disability, and school enrolment, and helping families access services, benefits, and support for their children [36].

## Discussion

Through the extensive community engagement and involvement (CEI) activities we developed a conceptual model for a community-based, cross-sectoral, family-centred inclusive intervention to prevent long-term health problems in children with ID by targeting modifiable risk factors from an early age (4–10 years), and address the stigma and social exclusion that exacerbate the course and experience of health problems. The intervention has the potential to address some of the modifiable risk factors by promoting a healthy lifestyle, preventing under and over nutrition, and improving healthcare access for children, and reducing stress and anxiety among the children and their parents in the short term. Although stakeholders did not mention the inclusion of children with disability into the mainstream health promotion activities, we added this to our conceptual model and Theory of Change to align with the sustainable and inclusive approach recommended by UNCRPD [37, 38].

Health promotion for children with ID can be implemented through the inclusive education programme. However, this will require attention to the structural, social and health system barriers, especially those related to stigma that limit inclusion. The conceptual framework aligns with the recommendation on health and early intervention for children with ID included in the India's National Policy for Persons with Disabilities (No.3-1/1993-DD.III, 2006), which promotes an inclusive approach. Our Theory of Change model outlines the pathways through which the intervention together with efforts to creating an enabling environment could translate into reducing communicable and non-communicable diseases, and mental health problems, thereby preventing morbidity and mortality in people with ID in India in the longer term. The framework can be adapted to other settings.

Wider activism for societal change, recognition of disability rights to an equal and inclusive world and economic support through social grants, etc will be required to address the socio-economic barriers to health. CEI being at the core of the work ensured acceptance and strong support from the community, from parents/carers of children with ID, adults with ID, educators, policy makers, clinicians, and civil society organisations.

People with ID are also socio-economically the most vulnerable population in India [12] and other LMICs [16] due to the limited opportunities for employment and income. Many have co-existing conditions (such as epilepsy) that require life-long treatment associated with extra expense for families. It is estimated that disability increases the cost of living by more than 30–40% of average income [39]. Thus, developing further morbidities (communicable and non-communicable diseases) would have far worse consequences for people with ID, than for others. Considering that healthcare spending in India is mostly out-of-pocket [40] and people with ID often have additional costs related to transportation and needing carers to

accompany them [2], developing long-term health problems would add to the burden of ongoing costs resulting in further impoverishment of this vulnerable population. Therefore, even a small effect of the intervention will have a large positive impact on individuals, families and the society in the long term. However, the conceptual model needs to be tested and further refined in order to develop an effective and sustainable intervention together with an enabling inclusive environment that can be scaled-up across India and also adapted by other LMICs. Although the alignment of the conceptual framework with India's National Policy for Persons with Disabilities will support integration [41], further inclusion within a wide range of existing government programmes would be needed to facilitate implementation, ownership and sustainability [42].

The conceptual model targets behaviour change, using the bio-psycho-social model and WHO's ICF [43], to support sustainable uptake. This will be further supported by the early provision of the intervention, plus an element of standardisation of simple to deliver core elements that maintain flexibility according to local circumstances. Research that tests cost-effectiveness and implementation by parents and educators is required [44]. Another important point in changing behaviour (in our case physical activity, diet, accessing healthcare) is to set expectations that are 'attainable' and 'reasonable' [44] within an enabling environment.

## Strengths and limitations

A community-based participatory approach was used for consulting and actively engaging stakeholders in co-developing a 'locally conceptualised' needs-based intervention to achieve meaningful results that maximises the benefits for children with ID and their parents. Our structured approach could be used to undertake other CEI activities in India and other LMICs, where communities and individual members of the public are not commonly engaged in developing an intervention. Consultations are thought to be more effective if they are conducted in open fora where the public can debate and challenge emerging concepts. Although we could not reach out to a wider community because of the social restrictions related to the pandemic, we minimised the chances of developing a biased model by listening to and evaluating multiple perspectives along with experts from the three national bodies who have several years of experience working with children and adults with ID in the Indian context. Another potential limitation is that systematic reviews used to inform the evidence base of the conceptual framework was from mainly high-income countries. Nevertheless, the evidence was a necessary starting point. Evidence generated from a LMIC context is required to strengthen the conceptual model, guide implementation and assess how best to align with the "disability-inclusive development" framework [45] when substantial socio-economic and health system challenges exist. Our work reinforces the United Nation's Convention on the Rights of Persons with Disabilities (UNCRPD) and the Sustainable Development Goals' (SDGs) commitment to "leave no one behind". It also contributes towards building inclusive societies and institutions and to the goals of 'Envision disability 2020' [46].

## Conclusion

There is an urgent need for appropriate strategies to reduce long-term health problems in children with ID as they move into earlier and later adulthood. Interventions guided by the conceptual framework have the potential to reduce and prevent long-term health problems in millions of children living with ID in India, and potentially a higher number globally. Considering that there are no health promotion programmes, which proactively include children with ID in LMICs, we have included a short guidance with key points to facilitate the use of the conceptual model (see S1 File). Further research is urgently required to test the conceptual

model in different settings, together with actions for improving the enabling environment in order to determine its acceptance and effectiveness.

## Supporting information

**S1 File. Guidance for using and contextually adapting the conceptual model.**
(DOCX)

**S1 Questionnaire. PLOS questionnaire on inclusivity in global research.**
(DOCX)

## Acknowledgments

We thank all stakeholders who participated in the CEI activities and shared their views, opinions, ideas, and reviewed the conceptual framework and the Theory of Change model. We also thank Professor Suresh Bada Math, Mr Arman Ali, Dr Shyamanta Das, Dr Mara Violato, Dr Binukumar Bhaskarapillai, Dr Louise Linsell and Professor John Vijay Sagar Kommu who were co-investigators for the wider project. They did not specifically contribute to the work included in this paper, but provided broader perspectives for developing the overall project.

## Author Contributions

**Conceptualization:** Manisha Nair, Mythili Hazarika, M Thomas Kishore, Nabarun Sengupta, Ganesh Sheregar, Hayley MacGregor, Mary Wickenden, Neel Harit Kaushik, Prarthana Saikia, Sasha Shepperd.

**Data curation:** Mythili Hazarika, Nabarun Sengupta.

**Formal analysis:** Manisha Nair, Mythili Hazarika, M Thomas Kishore, Nabarun Sengupta, Ganesh Sheregar, Neel Harit Kaushik, Prarthana Saikia, Sasha Shepperd.

**Funding acquisition:** Manisha Nair, Mythili Hazarika, M Thomas Kishore, Nabarun Sengupta, Ganesh Sheregar, Hayley MacGregor, Mary Wickenden, Maureen Kelley, Sasha Shepperd.

**Methodology:** Manisha Nair, M Thomas Kishore, Hayley MacGregor, Mary Wickenden, Maureen Kelley, Sasha Shepperd.

**Project administration:** Mythili Hazarika, Neel Harit Kaushik.

**Resources:** Manisha Nair, Mythili Hazarika, Ganesh Sheregar.

**Supervision:** Mythili Hazarika.

**Writing – original draft:** Manisha Nair.

**Writing – review & editing:** M Thomas Kishore, Nabarun Sengupta, Hayley MacGregor, Mary Wickenden, Maureen Kelley, Sasha Shepperd.

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
