## [Decision Letter · Decision Letter 0]

20 Sep 2022

PGPH-D-22-01074

A community developed conceptual model for reducing long-term health problems in children with intellectual disability in India

Dear Dr. Nair,

Thank you for submitting your manuscript to PLOS Global Public Health. After careful consideration, we feel that it has merit but does not fully meet PLOS Global Public Health’s publication criteria as it currently stands. Therefore, we invite you to submit a revised version of the manuscript that addresses the points raised during the review process.

We look forward to receiving your revised manuscript.

Kind regards,

Julia Robinson

Executive Editor

Journal Requirements:

2. Please include a complete copy of PLOS’ questionnaire on inclusivity in global research in your revised manuscript. Our policy for research in this area aims to improve transparency in the reporting of research performed outside of researchers’ own country or community. The policy applies to researchers who have travelled to a different country to conduct research, research with Indigenous populations or their lands, and research on cultural artefacts. The questionnaire can also be requested at the journal’s discretion for any other submissions, even if these conditions are not met.  Please find more information on the policy and a link to download a blank copy of the questionnaire here: https://journals.plos.org/globalpublichealth/s/best-practices-in-research-reporting. Please upload a completed version of your questionnaire as Supporting Information when you resubmit your manuscript.

3. Please insert an Ethics Statement at the beginning of your Methods section, under a subheading 'Ethics Statement'. It must include:

a) The name(s) of the Institutional Review Board(s) or Ethics Committee(s)

b) The approval number(s), or a statement that approval was granted by the named board(s) 

4. In the online submission form, you indicated that [Insert text from online submission form here]. All PLOS journals now require all data underlying the findings described in their manuscript to be freely available to other researchers, either 1. In a public repository, 2. Within the manuscript itself, or 3. Uploaded as supplementary information.

Additional Editor Comments (if provided):

Reviewers' comments:

Reviewer's Responses to Questions

**Comments to the Author**

1. Does this manuscript meet PLOS Global Public Health’s publication criteria? Is the manuscript technically sound, and do the data support the conclusions? The manuscript must describe methodologically and ethically rigorous research with conclusions that are appropriately drawn based on the data presented.

Reviewer #1: Yes

2. Has the statistical analysis been performed appropriately and rigorously?

Reviewer #1: N/A

3. Have the authors made all data underlying the findings in their manuscript fully available (please refer to the Data Availability Statement at the start of the manuscript PDF file)?

Reviewer #1: Yes

4. Is the manuscript presented in an intelligible fashion and written in standard English?

Reviewer #1: Yes

5. Review Comments to the Author

Reviewer #1: This project highlights the plight of a neglected section of society in one of the world's most populous countries. The authors have consulted widely and involved the consultees in reviewing the Theory of Change model they have jointly developed. The authors describe their conceptual framework to target the five evidence based core components for intervention. The Theory of Change model is not clearly described. Figure 2 is inevitably complex and crowded and the paragraph between lines 297 - 312 needs to be revised to highlight the components of the ToC.

The ToC mentions the "accountability ceiling" and the "Transfer of ownership" but there is no elaboration of the interface between local and national stakeholders. Without this bein clarified it is likely that the worthwhile issues put forward will not translate into policy and sustained action which this section of the population is crying out for

6. PLOS authors have the option to publish the peer review history of their article (what does this mean?). If published, this will include your full peer review and any attached files.

**Do you want your identity to be public for this peer review?** For information about this choice, including consent withdrawal, please see our Privacy Policy.

Reviewer #1: No

---

## [Decision Letter · Decision Letter 1]

30 Jan 2023

PGPH-D-22-01074R1

A community developed conceptual model for reducing long-term health problems in children with intellectual disability in India

Dear Authors,

Thank you for submitting your manuscript to PLOS Global Public Health. After careful consideration, we feel that it has merit but does not fully meet PLOS Global Public Health’s publication criteria as it currently stands. Therefore, we invite you to submit a revised version of the manuscript that addresses the points raised during the review process.

We look forward to receiving your revised manuscript.

Kind regards,

Shela Hirani, PhD, IBCLC, RN

Academic Editor

Journal Requirements:

2. We have noticed that you have uploaded Supporting Information files, but you have not included a list of legends. Please add a full list of legends for your Supporting Information files after the references list.

Additional Editor Comments (if provided):

Reviewers' comments:

Reviewer's Responses to Questions

**Comments to the Author**

1. If the authors have adequately addressed your comments raised in a previous round of review and you feel that this manuscript is now acceptable for publication, you may indicate that here to bypass the “Comments to the Author” section, enter your conflict of interest statement in the “Confidential to Editor” section, and submit your "Accept" recommendation.

Reviewer #1: All comments have been addressed

Reviewer #2: All comments have been addressed

2. Does this manuscript meet PLOS Global Public Health’s publication criteria? Is the manuscript technically sound, and do the data support the conclusions? The manuscript must describe methodologically and ethically rigorous research with conclusions that are appropriately drawn based on the data presented.

Reviewer #1: Yes

Reviewer #2: Partly

3. Has the statistical analysis been performed appropriately and rigorously?

Reviewer #1: N/A

Reviewer #2: N/A

4. Have the authors made all data underlying the findings in their manuscript fully available (please refer to the Data Availability Statement at the start of the manuscript PDF file)?

Reviewer #1: Yes

Reviewer #2: Yes

5. Is the manuscript presented in an intelligible fashion and written in standard English?

Reviewer #1: Yes

Reviewer #2: Yes

6. Review Comments to the Author

Reviewer #1: thank you responding to my earlier comments

Reviewer #2: The authors of this manuscript have done great job for their effort and energy put into this work. The subject matter of the manuscript is critical for the development and well-being of children with intellectual disability in India and other parts of the work where the conceptual framework and model can be replicated to develop and implement interventions for children with intellectual disability.

There is a clarification that I think needs to be made to strengthen the manuscript even more. That is the methodology section of this manuscript. Indeed, you have comprehensively described the consultative meetings and interviews with the various stakeholders. However, readers will still need to understand how these numerous stakeholders were sampled and what were the nature of the interviews and discussions conducted. Please, provide the inclusion and exclusion criteria for selecting various stakeholders from the institutions and communities.

The second issue I will like o see you improve is the diagram of conceptual framework. No doubt you did a wonderful job but it framework appears a little complex and needs to be simplified for the healthcare and educational instructors who will developing programs based on it. I will also recommend that you come out with some form of guidelines on how to use the model.

After reading the manuscripts many times, it sounds to me that the manuscript is a report of consultative discussions rather than a typical research article. This is more so you indicated this type of work is exempt from ethical review and approval. Please, clarify.

Despite the above, this work is undoubtedly comprehensive, fully participatory and well presented.

7. PLOS authors have the option to publish the peer review history of their article (what does this mean?). If published, this will include your full peer review and any attached files.

**Do you want your identity to be public for this peer review?** For information about this choice, including consent withdrawal, please see our Privacy Policy.

Reviewer #1: No

Reviewer #2: No

---

## [Decision Letter · Decision Letter 2]

27 Mar 2023

A community developed conceptual model for reducing long-term health problems in children with intellectual disability in India

PGPH-D-22-01074R2

Dear Authors,

We are pleased to inform you that your manuscript 'A community developed conceptual model for reducing long-term health problems in children with intellectual disability in India' has been provisionally accepted for publication in PLOS Global Public Health.

Best regards,

Shela Hirani, PhD, IBCLC, RN

Academic Editor

Reviewer Comments (if any, and for reference):

Reviewer's Responses to Questions

**Comments to the Author**

1. If the authors have adequately addressed your comments raised in a previous round of review and you feel that this manuscript is now acceptable for publication, you may indicate that here to bypass the “Comments to the Author” section, enter your conflict of interest statement in the “Confidential to Editor” section, and submit your "Accept" recommendation.

Reviewer #2: All comments have been addressed

2. Does this manuscript meet PLOS Global Public Health’s publication criteria? Is the manuscript technically sound, and do the data support the conclusions? The manuscript must describe methodologically and ethically rigorous research with conclusions that are appropriately drawn based on the data presented.

Reviewer #2: Yes

3. Has the statistical analysis been performed appropriately and rigorously?

Reviewer #2: N/A

4. Have the authors made all data underlying the findings in their manuscript fully available (please refer to the Data Availability Statement at the start of the manuscript PDF file)?

Reviewer #2: Yes

5. Is the manuscript presented in an intelligible fashion and written in standard English?

Reviewer #2: Yes

6. Review Comments to the Author

Reviewer #2: I will like to commend the authors for the great work they have done revising the manuscript. After several readings of the entire manuscript, I have observed that the manuscript is much improved after addressing all of the issues I raised in my earlier review. I believe that the time and effort taken to develop the conceptual framework that is evidenced-based for the reduction communicable and non-communicable disease infection among the special needs children will result in significant positive change in how the health of special needs children will be promoted in India and other parts of the world. This study will also not form the basis for similar innovations in health promotion for the mentally and intellectually challenged children but also a guide for program and intervention designers for the future.

7. PLOS authors have the option to publish the peer review history of their article (what does this mean?). If published, this will include your full peer review and any attached files.

**Do you want your identity to be public for this peer review?** For information about this choice, including consent withdrawal, please see our Privacy Policy.

Reviewer #2: No
